# The ER Stress/UPR Axis in Chronic Obstructive Pulmonary Disease and Idiopathic Pulmonary Fibrosis

**DOI:** 10.3390/life11010001

**Published:** 2020-12-22

**Authors:** Mahmoud Aghaei, Sanaz Dastghaib, Sajjad Aftabi, Mohamad-Reza Aghanoori, Javad Alizadeh, Pooneh Mokarram, Parvaneh Mehrbod, Milad Ashrafizadeh, Ali Zarrabi, Kielan Darcy McAlinden, Mathew Suji Eapen, Sukhwinder Singh Sohal, Pawan Sharma, Amir A. Zeki, Saeid Ghavami

**Affiliations:** 1Department of Human Anatomy and Cell Science, Rady Faculty of Health Sciences, Max Rady College of Medicine, University of Manitoba, Winnipeg, MB R3E 0J9, Canada; parsa16657@yahoo.com (M.A.); aftabis@myumanitoba.ca (S.A.); alizadej@myumanitoba.ca (J.A.); 2Department of Clinical Biochemistry, School of Pharmacy and Pharmaceutical Sciences, Isfahan University of Medical Sciences, Isfahan 8174673461, Iran; 3Department of Clinical Biochemistry, School of Medicine, Shiraz University of Medical Sciences, Shiraz 7134845794, Iran; suny.respina@gmail.com (S.D.); mokaram2@gmail.com (P.M.); 4Autophagy Research Center, Shiraz University of Medical Sciences, Shiraz 7134845794, Iran; 5Medical Physics Department, Cancer Care Manitoba, University of Manitoba, Winnipeg, MB R3E 0V9, Canada; 6Division of Neurodegenerative Disorders, St Boniface Hospital Albrechtsen Research Centre, University of Manitoba, Winnipeg, MB R2H 2A6, Canada; aghanoori@gmail.com; 7Department of Internal Medicine, University of Manitoba, Winnipeg, MB R3E 0V9, Canada; 8Research Institute of Oncology and Hematology, Cancer Care Manitoba, University of Manitoba, Winnipeg, MB R3E 0V9, Canada; 9Biology of Breathing Theme, Children Hospital Research Institute of Manitoba, University of Manitoba, Winnipeg, MB R3E 0V9, Canada; 10Influenza and Respiratory Viruses Department, Pasteur Institute of Iran, Tehran 1316943551, Iran; mehrbode@yahoo.com; 11Faculty of Engineering and Natural Sciences, Sabanci University, Orta Mahalle, Üniversite Caddesi No. 27, Orhanlı, Tuzla, 34956 Istanbul, Turkey; dvm.milad1994@gmail.com; 12Sabanci University Nanotechnology Research and Application Center (SUNUM), Tuzla, 34956 Istanbul, Turkey; alizarrabi@sabanciuniv.edu; 13Respiratory Translational Research Group, Department of Laboratory Medicine, School of Health Sciences, University of Tasmania, Launceston 7250, Tasmania, Australia; kielan.mcalinden@utas.edu.au (K.D.M.); mathew.eapen@utas.edu.au (M.S.E.); sssohal@utas.edu.au (S.S.S.); 14Center for Translational Medicine, Jane & Leonard Korman Respiratory Institute, Thomas Jefferson University, Philadelphia, PA 19107, USA; pawan.sharma@jefferson.edu; 15Davis School of Medicine, Department of Internal Medicine, Division of Pulmonary, Critical Care, and Sleep Medicine, UC Davis Lung Center, University of California, Davis, CA 95616, USA; aazeki@ucdavis.edu; 16Veterans Affairs Medical Center, Mather, CA 95655, USA

**Keywords:** endoplasmic reticulum, fibrosis, tissue remodeling, non-coding RNA, UPR, autophagy, lung disease

## Abstract

Cellular protein homeostasis in the lungs is constantly disrupted by recurrent exposure to various external and internal stressors, which may cause considerable protein secretion pressure on the endoplasmic reticulum (ER), resulting in the survival and differentiation of these cell types to meet the increased functional demands. Cells are able to induce a highly conserved adaptive mechanism, known as the unfolded protein response (UPR), to manage such stresses. UPR dysregulation and ER stress are involved in numerous human illnesses, such as metabolic syndrome, fibrotic diseases, and neurodegeneration, and cancer. Therefore, effective and specific compounds targeting the UPR pathway are being considered as potential therapies. This review focuses on the impact of both external and internal stressors on the ER in idiopathic pulmonary fibrosis (IPF) and chronic obstructive pulmonary disease (COPD) and discusses the role of the UPR signaling pathway activation in the control of cellular damage and specifically highlights the potential involvement of non-coding RNAs in COPD. Summaries of pathogenic mechanisms associated with the ER stress/UPR axis contributing to IPF and COPD, and promising pharmacological intervention strategies, are also presented.

## 1. Introduction

The endoplasmic reticulum (ER) is derived from the outer membrane of the nucleus and represents one of the most active intracellular organelles. In cells that have developed an ER, including those residing in the lungs, the balance between synthesis and protein secretion sometimes fails, or proteins cannot be properly degraded or reduced by proteases. A disturbance in the synthesis, secretion, or destruction of proteins may lead to the accumulation of non-folded or mis-folded proteins within the ER, disrupting its normal function [1,2]. Such an impairment of ER homeostasis is referred to as ER stress, which plays a significant role in cellular survival and function [3].

The unfolded protein response (UPR) signaling pathway in ER stress was first identified in Saccharomyces, in which ER stress is exclusively controlled by the serine/threonine-protein kinase/endoribonuclease inositol-requiring enzyme 1 (IRE1). This is in contrast to vertebrates, where three main signaling arms regulate UPR: IRE1, activating transcription factor (ATF) -6β, and protein kinase R like ER kinase (PERK) (Figure 1). These pathways transfer information on the protein folding status in the ER to the cytosol and nucleus to restore protein-folding capacity. Binding of the ER chaperone-binding immunoglobulin protein (BiP), also known as GRP-78 or heat shock 70 kDa protein 5 (HSPA5), to IRE1α causes its deactivation [4,5]. Under ER stress, BiP preferentially binds to mis-folded proteins and subsequently is released from PERK and IRE1, which initiates these proteins’ dimerization [6]. 

Recent studies demonstrated a role for ER stress in fibrosis in multiple organs, including the lungs [1,7]. For example, recent reports highlight the importance of ER stress and UPR in chronic obstructive pulmonary disease (COPD) [8,9,10], and the relation between ER stress and familial idiopathic pulmonary fibrosis (IPF) [7,11]. This review will focus on our current understanding of the role of ER stress and UPR in COPD and IPF. First, we will briefly review details of the UPR and the pathogeneses of COPD and IPF, after which we will discuss the potential link between the UPR and the onset, progression, and severity of these diseases.

## 2. Unfolded Protein Response

The UPR aims to reestablish ER homeostasis to ultimately facilitate cell adaptation to ER stress. Two main models for UPR activation have been discussed most frequently. The first model is similar to that of the heat shock response pathway and relies on the availability and regulated binding of the chaperone BiP [2,3,12,13]. BiP homeostatically binds to UPR sensor proteins to prevent their oligomerization, which is interrupted during ER stress. This action leads to the oligomerization of the monomers, resulting in the activation of UPR [14,15]. There is experimental evidence of the interaction between BiP and IRE1, ATF6, and PERK in unstressed cells with subsequent dissociation during ER stress [16,17,18,19,20]. “Filamentation induced by c-AMP” (FIC) domains are conserved domains from bacteria to eukaryotes including in humans and are involved in stress responses and infections [21,22]. They catalyze the transfer and covalent binding of adenosine monophosphate (AMP) (AMPylation/adenylation) to hydroxyl side chains of several proteins, such as Rho GTPases, which induces actin cytoskeleton collapse by inhibiting the interaction of Rho proteins with their downstream effectors. The AMPylation of BiP offers an additional regulatory mechanism affecting ER homeostasis. BiP activity serves as a quality control process for ER protein homeostasis to regulate the UPR [21,23]. FIC domains are not restricted to promoting the aforementioned post-translational modifications of GTPases, as they can reverse AMPylation and also catalyze deAMPylation of the BiP chaperone [21,22]. Both FIC and BiP are transcriptionally activated upon ER stress induction [23], confirming the role of FIC-mediated AMPylation of BiP in UPR [23]. Regulation of the UPR by FIC-mediated BiP AMPylation also protects photoreceptors in the *Drosophila* visual system in response to constant 72-hour light stress. It is required for the photoreceptor neurons to adapt to transient stress demands in order to maintain proper vision [24]. The human FIC-domain protein HYPE (or FICD) is also involved in AMPylation and deAMPylation of BiP [21,22,25]. A single Fic protein, called HYPE (Huntingtin yeast interacting protein), was shown to contain a tetratricopeptide repeat (TPR) that can be used as a cellular targeting sequence [22]. It uses the same active site on BiP (threonine 518) to affect both AMPylation and deAMPylation in a time- and concentration-dependent manner [26]. When AMPylation activity of FICD is inhibited, the default activity is deAMPylation [22,25]. It has been proposed that the dimerization of FIC inhibits its AMPylation without affecting the deAMPylation activity [26]. This process requires a functional switch in FICD to change the unfolded protein load; yet its molecular basis remains unclear. 

Dysregulation of BiP is associated with numerous diseases, including neurological disorders and cancers [23]. BiP AMPylation is increased in a stable situation, when unfolded proteins are reduced, but decreased during ER stress. AMPylation locks BiP into a state resembling the ATP-bound conformation with high “substrate off” rates, thereby inhibiting its chaperone function [24]. BiP in its ATP-bound conformation is the favored substrate for AMPylation [25]. BIP AMPylation regulates UPR by chaperone inhibition to match the concentration of active chaperone to the amount of unfolded proteins [25]. A disturbance of the BiP AMPylation/deAMPylation cycle not only affects the ability of BiP to respond to misfolded protein aggregates but also interferes with its regulation of UPR activation [24].

The currently accepted model identifies that misfolded or unfolded proteins bind directly to UPR sensor proteins (including PERK and IRE1), which drives their oligomerization and subsequent activation of UPR [27,28,29]. One study suggests that activation via this model requires two steps: the separation of BiP from IRE1, resulting in cluster formation, followed by a direct interaction of unfolded proteins with the core stress-sensing region of the dimerized IRE1 cluster, rendering highly oligomerized and active IRE1 [27,30]. 

It is debatable which of these models is most accurate as the basal cellular abundance of UPR sensor proteins is relatively low and the measurement of formed complexes during ER stress is challenging [31]. Emerging evidence supports the second model for UPR activation as emerging new data seem to indicate that preformed UPR sensor complexes are activated upon the direct binding of misfolded proteins [32]. Under conditions of stress caused by the accumulation of misfolded or aggregated proteins, UPR is regulated and activated by IRE1 via non-conventional splicing; translationally by PERK through phosphorylation of eukaryotic translation initiation factor 2A (eIF2a); and by ATF6 via regulated proteolysis [14]. BiP is also recruited to the ER lumen to increase its folding capacity as the load of accumulated misfolded protein becomes greater and the dissociation of BiP from the three UPR sensors further increases their activation [33]. Activated ATF6 plays a role in this process. Upon the accumulation of misfolded proteins, ATF6 is translocated to the Golgi apparatus and processed by the proteases Site 1 Protease (S1P) and Site 2 Protease (S2P); this produces a cleaved cytosolic ATF6 fragment that acts as a pivotal transcriptional factor to induce genes encoding BiP and several other chaperone proteins that mediate ER expansion and increase ER folding capacity [1,33]. 

### 2.1. Integral UPR Pathway Proteins

The three UPR sensors are not activated simultaneously. Activation of ATF6 and IRE1 is immediate and gradually attenuates, whereas PERK activation occurs later and can persist with chronic ER stress [34,35]. The activation of both ATF6 and IRE1 induces several transcriptional pathways dedicated to developing ER protein folding capacity and promoting selective protein degradation [1] (Figure 1).

#### 2.1.1. IRE1 

Two human isoforms of IRE1, IRE1α and IRE1β, have thus far been identified encoded by *ERN1* and *ERN2*, respectively. IRE1α is expressed ubiquitously, while IRE1β is predominantly expressed in goblet cells of the intestinal epithelium and the lungs [36]. Interestingly, IRE1α−/− mice are embryonically lethal, whereas IRE1β−/− mice are viable [37,38]. Furthermore, IRE1β−/− mice display a gross accumulation of Mucin 2 (MUC2) and intense ER stress, revealing that (in goblet cells) IRE1β negatively affects MUC2 expression and that an unsupported stockpile of MUC2 in the ER produces ER stress [38]. IRE1α has been the focus of major research as it was initially identified as having a vital role in the determination of cell fate, but we now know it is also intricately linked to ER stress and UPR signaling [39,40]. IRE1α is a type 1 ER transmembrane protein possessing both intrinsic kinase and endoribonuclease activity [19,29,36]. IRE1α can initiate UPR signaling through its endoribonuclease activity and subsequent unconventional selective splicing of a segment of XBP1 mRNA, which is localized to the ER, allowing the translation of active XBP1 [41,42,43,44,45] (Figure 1). The transcription factor XBP1 then enters the nucleus and upregulates several UPR genes encoding ER chaperones and activates the UPR element (UPRE), which is critical for the regulation of the ER-associated degradation (ERAD) system [46]. The genes regulated by the interaction between IRE1α and XBP1 ultimately enhance protein transport and folding in the ER and also promote relevant degradation pathways [47]. At least four molecules make up IRE1α oligomers. These oligomers dissociate with chronic ER stress, resulting in a reduction in IRE1 endoribonuclease activity and dephosphorylation of the kinase [48]. Autophosphorylation of IRE1α at the ser724 residue has been observed during the initiation of the UPR signaling cascade [49]. In addition, IRE1α can induce UPR signaling through post-transcriptional modifications, such as ERAD of RNAs via Regulated IRE1-Dependent Decay (RIDD), to reduce the ER load [50,51].

IRE1α is most active in the early stages of ER stress; following its autophosphorylation, IRE1α kinase and ribonuclease activities are initiated. The latter cleaves XBP1u mRNA to produce a mRNA expressing an active transcription factor called spliced XBP1 (XBP1s). The XBP1s protein controls the transcription of genes involved in ERAD, protein folding, and phospholipid synthesis [52,53]. The endoribonuclease also degrades specific mRNAs, identified as the RIDD process, and IRE1α kinase activity generates stress alert pathways by binding to TNF receptor-associated factor 2 (TRAF2) adapter proteins, resulting in nuclear factor-kappa B (NF-kB)- and Jun NH2-terminal kinase (JNK)-mediated signaling [54].

Inactivating oligomerization and AMPylation decreases BiP expression [22,25]. For years, this process has proven relevant for the recovery of BiP from the complex with IRE1 under ER stress conditions [16,17,55]. A recent study demonstrated an ATP- and co-chaperone-dependent system by which BiP stimulates the formation of a monomeric, inactive state of IRE1’s stress-sensing luminal domain (IRE1LD), which further supports the existence of a competitive suppression model in which the potential of BIP to affect key conversions in IRE1 activity is modified [56]. Amin-Wetzel et al. (2019) reported that compelled loading of the endogenous BiP onto endogenous IRE1α suppressed UPR signaling in CHO cells. Deletions in the IRE1α locus that de-suppressed the UPR encrypted flexible regions of IRE1LD mediate BiP-induced monomerization in vitro [56].

#### 2.1.2. ATF6

ATF6 is a type 2 transmembrane receptor and ER stress-regulated transcription factor. ATF6 activates the transcription and expression of an array of components important in boosting protein folding, regulating protein degradation, and ER expansion [57]. As discussed earlier, the bond between BiP and ATF6 is relieved during ER stress, allowing the translocation of these proteins to the Golgi apparatus [58], where the cytosolic domain of ATF6 is cleaved by S1P and S2P to produce an active transcription factor [15,59,60]. The transcriptional upregulation of XBP1 mRNA, which is non-canonically spliced by IRE1α, is also mediated by activated ATF6, further allowing the translation and activation of XBP1 [57]. In addition, ATF6 can regulate ER stress by binding to and inducing transcription through ER stress response element (ERSE) and via the activation of multiple ER chaperone proteins [46,61,62] (Figure 1). 

#### 2.1.3. PERK

PERK is a type 1 ER transmembrane protein (like IRE1), encoded by *EIF2AK3,* and has intrinsic kinase activity. PERK was initially identified as an integral protein in the UPR pathway and has been recognized as a key player in insulin biogenesis. PERK deficiency has been demonstrated to cause Wolcott Rallison syndrome (WRS), leading to permanent neonatal diabetes and the onset of growth retardation later in life [63,64]. In the ER lumen, PERK is kept inactive through binding to BiP; however, the cytoplasmic domain of PERK harbors Ser/Thr kinase activity, and PERK activation is followed by autophosphorylation of its kinase domain, providing PERK with full catalytic activity [36]. Upon the accumulation of misfolded proteins and initiation of ER stress, BiP bound to PERK is released, allowing for dimerization, autophosphorylation, and activation. PERK then selectively phosphorylates the α-subunit of eIF2α, leading to the translational attenuation of several mRNAs, reducing the load on the ER and inhibiting the translation of a multitude of proteins involved in cell growth [65,66,67].

On the other hand, phosphorylation of elF2α also favors the translation of ATF4 mRNA along with other selective mRNAs during the activation of stress-responsive transcription factors [66,68]. In addition, PERK induces phosphorylation of the transcription factor nuclear factor erythroid 2-related factor 2 (NRF2), promoting its translocation to the nucleus [69]. Activated NRF2 is required for free radical scavenging and controlling redox homeostasis [70]. In response to ER stress, NRF2 works synergistically with ATF4 to activate genes associated with the modulation of redox homeostasis [71,72,73]. PERK signaling also controls pathways that regulate the status of mitochondria during ER stress by preventing the accumulation of proteins that may affect mitochondrial function [74]. Furthermore, PERK-dependent regulation of stress-induced mitochondrial hyperfusion (SIMH), a survival mechanism in which protective elongation of mitochondria occurs along with increased production of ATP [75,76,77], serves as a mitochondrial protective mechanism during ER stress and constitutes another facet of the UPR pathway. A schematic summary highlighting the main signaling arms of the UPR as a communication mechanism between the ER and nucleus is shown in Figure 1. 

In summary, misfolded proteins bind directly to UPR sensor proteins at the same time as BiP dissociates; this drives sensor oligomerization and results in the activation of UPR [14,78,79]. Activated proteins of the UPR pathway play key roles in the regulation of protein folding, ER expansion, and pro-survival responses. Conversely, UPR-mediated signaling is also importantly involved in other cellular processes, such as apoptosis, autophagy, and pro-death responses [67,78,80].

### 2.2. Signaling Pathways Activated by UPR 

Under conditions where the UPR is hindered and the load of unfolded proteins becomes higher, the UPR can initiate apoptosis [81], autophagy [2,36,82,83], or cell death [33,44,45]. Although the ER is quite robust and resilient, often, cells are operating at the limits of their capacity [84]. With continued stress, when the cell fails to restore ER homeostasis and appropriate protein folding, the UPR adopts alternate signaling termed “terminal UPR”, which pushes the cell towards apoptosis [85].

IRE1α is required for ER stress-induced activation of apoptosis, whereas PERK and ATF6 can also be involved but are not essential [39]. IRE1α has been linked to a regulatory role in ER stress-induced apoptosis as it forms a complex with apoptosis signal-regulating kinase 1 (ASK1) and activates its downstream target c-Jun NH2-terminal kinase (JNK) via binding to TNF receptor-associated factor 2 (TRAF2) [86,87,88,89]. Excessive activation of PERK results in the upregulation of the growth arrest and DNA damage-inducible gene 153 C/EBP homologous protein (CHOP or GADD153), a transcription factor that inhibits the encoding of the anti-apoptotic B-cell lymphoma (BCL) 2 gene and promotes the expression of related pro-apoptotic proteins [90]. Furthermore, PERK is a critical part of mitochondrial associated membranes (MAMs), which are important in sustaining the structural and functional integrity of mitochondria, maintaining calcium dynamics, and regulating metabolism as a communication mechanism between the ER and mitochondria [77,91]. The absence of PERK disrupts MAM-associated mechanisms, resulting in ER fragmentation and a reduction in apoptosis induced by reactive oxygen species (ROS) and mediated by ER stress with perturbed Ca^2+^ signaling [77]. Hence, PERK acts as an interface between ER and mitochondria, regulating ROS-induced apoptosis, and the lack of PERK protects mitochondria from ROS. 

Autophagy is another homeostatic process that is increasingly activated upon changes in the cell, such as starvation, hypoxia, or ER stress [80,92,93,94,95]. Autophagy is a recycling mechanism that degrades aggregated proteins and toxic components, relieving cellular stress and maintaining homeostasis [94,96,97,98,99]. Autophagy flux is increased with persistent ER stress to promote cell survival. ATF4 and CHOP (both activated by PERK) are mostly associated with cellular death pathways upon overexpression; however, autophagy is transcriptionally regulated by both ATF4 and CHOP and can oppose terminal UPR [33,100]. These two proteins have also been identified in the regulation of numerous autophagy-related genes (ATG) [101]. ER stress also activates JNK via the interaction between IRE1α and TRAF2, which ultimately phosphorylates Bcl2 and leads to dissociation of Bcl2 and Beclin1 proteins, enabling the activation of the phosphoinositide-3-kinase (PI3K) complex and initiation of autophagy [102]. Following continued cellular stress, JNK-mediated autophagy can occur independently of IRE1 [103]. Autophagy deficiency and inhibition renders UPR-dependent inflammation in intestinal epithelial cells, highlighting another functional association between UPR and autophagy with pathophysiological implications [104].

An additional route of ER stress-induced apoptosis is via a caspase-dependent pathway. Caspase-12 belongs to the group 1 family or IL-1β-converting enzyme (ICE)-like caspases and is activated following ER stress, mediating apoptosis independently of mitochondria [105,106,107]. ER stress leads to the translocation of Bim, a BH-3 (Bcl-2 homology domain-3)-only protein, to the ER, resulting in caspase-12 activation [108]. Under conditions of ER stress, caspase-12 directly activates caspase-9, and it has been proposed that caspase-3 is activated downstream in this cascade, triggering apoptosis [109,110]. Selective activation of caspase-12 has been demonstrated in the motor neurons of rabbits following spinal cord trauma [111]. In addition, caspase-12 has been reported to be activated as a result of treatment with two chemotherapy drugs [105,112]. 

CHOP also plays an important role downstream in the pro-death response [45,113]. Whilst ATF4 can activate CHOP, IRE1α can lead to the phosphorylation and activation of CHOP through binding and activation of p38 MAPK [57]. CHOP downregulates Bcl-2 and upregulates the transcription of Bim, ultimately resulting in the downstream initiation of an apoptotic cascade [114,115,116]. Interestingly, reduced levels of apoptosis in response to ER stress have been observed in CHOP knockout mice [117]. 

## 3. Chronic Obstructive Pulmonary Disease

Chronic obstructive pulmonary disease (COPD) is the third leading cause of death and fifth most prevalent disease worldwide [118,119]. COPD is a chronic and progressive lung disease characterized by irreversible airway obstruction due to obliteration of the small airways and loss of alveoli, which leads to emphysema. Some COPD patients can also manifest airway hyperreactivity associated with smoking and/or concurrent allergic disease. Symptoms include chronic cough, sputum production, dyspnea, intermittent wheezing, chronic airflow obstruction, and reduced to poor quality of life [120,121]. Although the underlying mechanisms of COPD are not completely understood, the disease is also associated with chronic corticosteroid-resistant inflammation, which contributes to the limited therapeutic options that are currently available. 

### 3.1. COPD Pathophysiology

COPD is defined by chronic airway inflammation and enlargement of mucous glands in the central airways. This leads to mucus hypersecretion and airway mucus plugging, epithelial injury, airway fibrosis, and peripheral airway damage, including emphysema (i.e., alveolar destruction and airspace enlargement) that begins in the respiratory bronchioles and ends in the lung parenchyma [122]. The Global Initiative for Chronic Obstructive Lung Disease (GOLD) classifies and stages COPD according to the severity of airflow obstruction based on spirometric measures (GOLD 1–4). In addition to COPD severity stage based on lung function, patients are also classified according to their symptoms, functional limitations, and number and severity of acute exacerbations which places them in one of four Groups: A-D (COPD pathology is summarized in Figure 2). 

### 3.2. Onset and Progression of COPD: The Role of Cigarette Smoke

While various environmental and genetic risk factors have been identified in COPD, most cases of COPD are due to long-term cigarette smoke (CS) exposure. Yet, only about 15% of all smokers develop COPD in their lifetime [123]. Exposure to CS induces oxidative stress, which in turn triggers alveolar macrophages to generate ROS or reactive nitrogen species (RNS). These reactive compounds cause cytokine and chemokine release, leading to the infiltration of lymphocytes and neutrophils into airways and lung tissue. Interferon-γ (IFN-γ) is released by both helper and cytotoxic lymphocytes, while innate neutrophils produce neutrophil elastase that causes mucus hypersecretion and local tissue damage. 

CS also induces NF-κB by inhibiting histone deacetylase 2 (HDAC2), thereby promoting lung inflammation [124,125]. In addition, oxidative stress activates the PI3K pathway, which phosphorylates and inactivates HDAC2, leading to ubiquitinated proteasomal degradation and a subsequent increase in the expression of pro-inflammatory genes, ultimately resulting in impaired corticosteroid responsiveness [126]. 

Epithelial cells and macrophages release proteases, such as matrix metalloproteinase-9 (MMP-9), which contributes to elastin degradation and the development of emphysema. These cells also release transforming growth factor-β (TGF-β) and fibroblast growth factor (FGF), which stimulate fibroblast proliferation and are importantly involved in the pro-fibrotic response. In addition, epithelial growth factor (EGF) and TGF-α stimulate mucus hypersecretion [127], which is further exacerbated by inflammation.

### 3.3. COPD and UPR: Cellular Crosstalk Mechanisms 

#### 3.3.1. Oxidative Stress in COPD

Airway cell damage in COPD is largely attributed to oxidative and carbonyl stress [128,129]. Oxidative stress occurs when free radical exposure is able to suppress antioxidant defenses. Ubiquitous free radicals (e.g. ROS) primarily arise during mitochondrial respiration and/or pathogen restriction responses. ROS species, such as hydroxyl radical (**·**OH) and superoxide anion (O2**·**^−^), contain unpaired electrons which they rapidly transfer to other molecules through oxidation, resulting in damage to proteins, lipids, and/or DNA and potentially leading to the generation of novel ROS [130] (Figure 2). Since the protein transport and folding processes are ATP (energy)-demanding and generate ROS, UPR induces the transcription of genes that modulate energy synthesis and ROS quenching [131]. Of note, many basic cellular functions, including cell cycle regulation, apoptosis, energy metabolism, inflammation, and acute phase reactions that depend on an adequate supply of fully functional ER membranes and secreted proteins, are also regulated by the UPR [34,132]. As stated earlier, UPR is controlled by three ER-transmembrane stress sensors (IRE1α, PERK, and ATF6) [133,134,135]. Under normal physiological circumstances, activation of these sensors is inhibited by the binding of their luminal domains to the main and most represented ER-resident chaperone BiP. However, following CS exposure and resultant ROS production, the UPR pathway becomes activated. 

#### 3.3.2. CS-Induced UPR Signaling in COPD

Apart from nicotine, heavy metals, and over 4000 chemicals, tobacco smoke exposes airway tissue to high concentrations of oxidants and free radicals. For instance, a single puff of CS contains an excess of 1 × 10^15^ oxidant molecules that could impact lung health [136,137]. Oxidants, including ROS and RNS, mediate oxidative stress responses that overwhelm the protective antioxidant defenses. In addition to potentiating inflammatory processes, smoking-associated damage of lung resident macromolecules can also sustain tissue injury and cell death through modification of amino acids and proteins and peroxidation of lipids [138,139,140]. 

In both human COPD lungs and those of CS-exposed mice, there is evidence of increased levels of insoluble, poly-ubiquitinated, high-molecular-weight proteins [141,142]. Murine studies have shown that CS exposure increases the levels of misfolded proteins, including functionally impaired protein disulfide isomerase (PDI), a critical foldase that modulates ER function [143,144]. In humans, CS-induced oxidative stress causes irreversible damage to lung protein structures in both airway and alveolar epithelial cells, which leads to degradation by the ubiquitin–proteasome system or via autophagic vacuoles [144,145,146]. Concomitantly, CS exposure alters the activity of proteasome constituent trypsin, chymotrypsin, and caspases, thereby affecting the elimination of misfolded proteins [145,147] (Figure 2). 

Although the implied role of UPR in COPD is largely accepted, the precise downstream interactions are ambiguous and perceived to be biologically diverse. Numerous major proteins of the UPR pathway have been implicated and have the potential to be involved in inflammation [148,149]. This has fueled research into the correlations between portions or entire arms of the UPR pathway and COPD pathogenesis. ER proteins with significant roles in UPR, such as BiP, may be upregulated in bronchoalveolar lavage (BAL) fluid and lung samples taken from smokers [150,151]. While expression levels of phospho-eIF2α and CHOP correlate with the severity of airway obstruction, such increments are largely independent of smoking and associated with stress-induced increases in caspase-3 and -7 in COPD lungs [152]. Independent of smoking history, diminished miR199a-5p expression within peripheral blood monocytes (PBMCs) derived from COPD patients correlates with an increase in UPR markers, such as BIP, ATF6, and sXBP1 [153]. The increased presence of such ER stress/UPR markers in COPD patients could be correlated with adverse airway remodeling; importantly, these proteins could prove to be useful biomarkers and represent potential therapeutic targets [154]. These findings contradict observations made in alveolar type II epithelial cells derived from COPD lungs, where only modest expression of BiP, sXBP1, CHOP, and ATF6 was detected as compared to IPF [155].

#### 3.3.3. UPR and Autophagy Signaling in COPD

Autophagy is widely recognized as a key regulator of innate and adaptive immune mechanisms that profoundly impacts disease pathogenesis [156,157,158,159,160]. Important processes affected by autophagy include inflammation, antigen presentation, viral infection, and bacterial clearance [157].

COPD patients show a significant elevation in autophagic proteins, with increased levels of LC3B-II, a marker of autophagosome formation. This is accompanied by a significant increase in other autophagy-associated proteins, such as ATG4, ATG5–ATG12, and ATG7, in COPD as compared to normal lungs [161]. While CS is known to further enhance the levels of autophagic markers in vitro and in vivo, increased autophagy has also been observed in a genetic variant of emphysema, α1-anti-trypsin (A1AT) deficiency, in which the etiology can be viewed as independent of smoke or noxious particle inhalation [162,163].

Under nutrient-rich conditions, autophagy is regulated by class I PI3K and mammalian target of rapamycin (mTOR) and is susceptible to activation by UPR [164]. While the UPR and mTOR complex 1 (mTORC1) mediate autophagy bidirectionally, activation of the PERK-eIF2α axis is critical for autophagy activity that is associated with ER stress [100,101]. PERK also stimulates autophagy by inhibiting Akt- and ATF4-mediated induction of autophagy-associated genes [165]. 

#### 3.3.4. UPR-Associated Diagnostic Markers in COPD

Controversies exist regarding the role of UPR in COPD. Increased UPR activity has been postulated to contribute to lung cell apoptosis. Conversely, diminished UPR activity has been proposed to explain increased levels of misfolded protein aggregates and impaired antioxidant defenses [141,152]. These functional contradictions could be due to genetic propensity or phenotypic diversity as UPR gene expression considerably varies among individuals and is modulated by specific stimuli [166]. Additional research is needed in this area. 

#### 3.3.5. Non-Coding RNAs and UPR in COPD

Non-coding RNAs (ncRNAs) are a category of RNA molecules that do not contribute to protein synthesis [167]. More than 90% of RNAs made from the human genome are ncRNAs and a high number of ncRNAs have been recognized in recent years [168,169,170]. Notably, ncRNAs have been shown to participate in physiological and pathological events. Physiological processes, such as apoptosis, proliferation, differentiation, migration, and autophagy, are under close surveillance by ncRNAs. In addition, ncRNAs play a key role in the development of several pathological conditions, including cancer, diabetes, cardiovascular diseases, and other disorders [171]. ncRNAs can be used as reliable biomarkers for disease diagnosis, as some ncRNAs can be measured depending on their stability in the bloodstream [172,173]. Furthermore, it is worth mentioning that ncRNAs can be therapeutically targeted by pharmacological or genetic interventions [174]. Considering these features, ncRNAs represent potential diagnostic, prognostic, and therapeutic options in disease [175].

Recent studies have highlighted the involvement of ncRNAs in COPD development and the regulation of disease-associated molecular mechanisms. For instance, expression of microRNA (miRNA)-155 is increased in smokers and COPD patients compared to never-smokers [176]. In addition, miR155-deficient mice show an attenuation of the CS-induced increase in inflammation-related genes, and intranasal instillation of a specific inhibitor of miRNA-155 inhibited CS-induced inflammation in wild-type mice [176]. Similarly, miRNA-29b appears to regulate pathogenesis in a rat model of COPD [177]. 

In addition to in vitro and in vivo experiments, clinical studies have also shown a role of miRNAs in COPD. Female COPD patients smoking tobacco were compared to female COPD patients who never smoked but were exposed to biomass smoke (BS). miRNA-22-3p were lower in the BS compared to the tobacco smoking group, thus these findings would suggest a pro-inflammatory role for miR02203p [178]. Other kinds of ncRNAs, including long non-coding (lncRNAs) and circular (circRNAs) RNAs, have also been implicated in COPD pathogenesis [169,179,180,181]. Interestingly, we now know that ncRNAs can also affect the UPR [182,183]. To date, just one study has evaluated the role of ncRNAs in the regulation of UPR in COPD, by assessing and analyzing miRNA expression profiles in peripheral blood monocytes collected from patients. During ER stress, expression of miRNA-199a-5p is increased. Results demonstrated that there was an association between miRNA-199a-5p and miRNA-199a-2; miRNA-199a-2 promoter hypermethylation prevented miRNA-199a-5p expression. In addition, BiP and ATF6 were subject to regulation by miRNA-199a-5p in COPD monocytes. MiRNA-199a-5p binds to the 3/-untranslated region (3/-UTR) of BiP and ATF6 to reduce their expression. The other arms of the UPR, IRE1 and PERK, were also downregulated by miRNA-199a-5p in COPD [153]. Epigenetic regulation of this miRNA is important in modulating the expression of the UPR arms. Moreover, the findings of this study provide insight into novel therapeutic strategies for COPD. However, more studies are needed to further identify specific regulatory roles of ncRNAs, including miRNAs, circRNAs, and lncRNAs, in the UPR in COPD patients.

## 4. Idiopathic Pulmonary Fibrosis (IPF)

IPF is a chronic, progressive parenchymal lung disease that is pathologically identified as usual interstitial pneumonia (UIP) with unknown etiology, occurring primarily in adults (usually >50 years old) [184,185,186]. Familial pulmonary fibrosis (FPF) or familial interstitial pneumonia (FIP) is defined as two or more members within the same family having IPF or any other form of idiopathic interstitial pneumonia (IIP). Approximately 0.5–3.7% of IPF cases are familial in origin [187,188,189,190,191,192,193]. 

Interestingly, 2–20% of IPF patients have a first-degree relative with interstitial lung disease (ILD). The estimated incidence range is higher in Europe and North America and lower in East Asia and South America. Men are affected more than women by a ratio of nearly 2:1, which progressively increases with age [194]. The median life expectancy in IPF ranges from 2 to 5 years after initial diagnosis, which is comparable to that of many cancers (e.g., breast, ovarian, and colorectal cancer) [195,196,197,198]. Therefore, IPF has been recognized as a major life-threatening pulmonary disease and is usually progressive and lethal [199,200]. 

The clinical symptoms of IPF include unexplained chronic exertional dyspnea along with a frequent non-productive cough, fine bibasilar inspiratory crackles that are often “high pitched” or “velcro-like” in character, and finger nail clubbing [201,202,203,204]. Potential risk factors include cigarette smoking, environmental and occupational exposures (i.e., metal dust, wood dust, sand, stone, silica, farming, and livestock), microbial agents, gastroesophageal reflux, diabetes mellitus, and various genetic factors (both familial and sporadic polymorphisms) [184,205,206,207,208,209].

### 4.1. IPF Pathophysiology

IPF is also known as cryptogenic fibrosing alveolitis, a name less-commonly used, and differs from other classes of pulmonary fibrosis, such as desquamative interstitial pneumonia (DIP), acute interstitial pneumonia (AIP), nonspecific interstitial pneumonia (NSIP), and cryptogenic organizing pneumonia (COP). IPF usually demonstrates a histopathologic UIP pattern diagnosed by lung biopsy [195,210]. The key histopathological features of UIP/IPF on lung biopsy include a combination of (1) irregularly distributed fibrosis along with scarring (consists of dense acellular collagen deposition), (2) heterogeneous patchy interstitial infiltrates of lymphocytes and plasma cells with subpleural and para-septal lung parenchymal mild inflammation and hyperplasia, (3) temporal heterogeneity of fibrosis characterized by scattered fibroblast foci (convex subepithelial foci of proliferating fibroblasts and myofibroblasts), and (4) honeycomb change (consisting of cystic fibrotic airspaces) [211,212,213,214,215,216,217].

Although surgical biopsy remains the gold standard tool for diagnosing the UIP pattern, high-resolution computed tomography (HRCT) serves as an essential component for demonstrating radiological images that correlate with UIP in patients with IPF [218,219]. The typical HRCT findings that correlate with a histopathological pattern of UIP include the presence of reticular opacities (often associated with traction bronchiectasis), honeycomb change (manifested as clustered cystic airspaces of 3–10 mm diameter), and a patchy basal and peripheral distribution of fibrosis [220,221,222,223,224]. An international evidence-based consensus established jointly by the American Thoracic Society (ATS), the European Respiratory Society (ERS), the Japanese Respiratory Society (JRS), and the Latin American Thoracic Association (ALAT) defines IPF diagnostic criteria as follows: the presence of UIP pattern on HRCT in patients without lung biopsy, UIP pattern noted on combined HRCT and lung biopsy, and the exclusion of other known causes of ILD (e.g., domestic and occupational environmental exposures, connective tissue disease, and drug toxicity) [184,225,226] (Figure 3).

### 4.2. Onset and Progression of IPF

The cause of IPF is unknown; however, many clinical and environmental exposure associations exist. Prior to the last decade, the pathogenesis of IPF was very poorly understood, but over recent years, two different hypotheses for the pathogenesis of IPF have been proposed. The traditional “inflammation/alveolitis” hypothesis states that IPF is due to chronic inflammation, occurring in response to an unknown stimulus (or stimuli). If this inflammation is left untreated, it will lead to progressive lung injuries and fibrosis [211,227]. However, anti-inflammatory and immunosuppressive therapies (i.e., oral corticosteroids and cytotoxic agents) have failed to significantly improve disease progression or its prognosis. Indeed, some animal models demonstrate pulmonary fibrosis in the absence of inflammation [228,229]. An alternative hypothesis emphasizes non-inflammatory mechanisms, such as epithelial-mesenchymal transition (EMT), and states that repeated, unidentified, exogenous, and/or endogenous stimuli trigger the development of IPF. These stimuli lead to sequential microscopic lung injuries, with noticeable disruption in the integrity of the alveolar epithelium, that induce an abnormal wound healing process characterized by an uncontrolled proliferation of myofibroblasts. The disorganized proliferation of myofibroblasts results in the gradual distortion of the epithelial lining. This leads to the formation of fibroblast-myofibroblast foci and, consequently, lung fibrosis, a histological hallmark of IPF [227,230,231,232]. Thus, in this “epithelial/mesenchymal” hypothesis, pulmonary fibrosis is the final pathological outcome of aberrant wound healing that is activated in response to unknown persistent or recurrent lung injury. Accordingly, abnormal lung wound repair has gained significant interest regarding the pathogenesis of IPF. This process comprises complex and multistage physiologic mechanisms that include collagen production, angiogenesis, cell migration, and cellular proliferation triggered by unknown tissue injuries [233]. 

In IPF, type I alveolar epithelial cells (AECs-I) are typically damaged and shed off the alveolar lining layer, possibly due to enhanced apoptosis, aging/shortening of telomeres, and activation of stress response pathways. Moreover, deficiencies in the normal re-epithelialization process force restoration of the epithelial surface by type II alveolar epithelial cells (AECs-II). This abnormal restoration attempt results in epithelial hyperplasia and triggers the secretion of several profibrotic factors, such as tissue factor plasminogen activator inhibitor (PAI)-1 and -2, by injured/activated AECs. These pro-fibrotic factors create a procoagulant/anti-fibrinolytic intra-alveolar environment that facilitates increased fibrotic responses [234,235,236]. Subsequently, this stimulates the secretion of multiple cytokines and growth factors, including TGF-β, TNF-α, and PDGF. Such cytokines promote the disruption of the epithelial basement membrane and increase the migration, proliferation, and accumulation of alveolar fibroblasts and myofibroblasts, which are relatively resistant to apoptosis. Accumulation of these cells increases the extracellular matrix (ECM) that results in the formation of fibroblast foci, a typical characteristic of the UIP pattern and morphological feature in IPF. Fibroblasts and myofibroblasts secrete fibrillar collagens, fibronectin, elastic fibers, and prostaglandins (PGs), further contributing to ECM deposition, and play a major role in lung parenchymal ECM remodeling, which represents another hallmark of aberrant tissue remodeling in IPF [237,238]. ECM remodeling, dysregulation of lung architecture, and the amount of fibroblast/myofibroblasts foci are considered major prognostic factors in IPF patients [201].

Squamous cell carcinoma antigen (SCCA), which is a marker for epithelial instability and/or EMT, is overexpressed in IPF [239,240,241]. EMT is the process by which epithelial cells lose their typical characteristics (such as cuboidal shape, apical-basal polarization, cell–cell contacts, epithelial gene repertoire, and downregulation of epithelial marker E-cadherin) and acquire mesenchymal cell-like features (e.g., spindle morphology, loss of cell contacts, mesenchymal gene expression, and upregulation of mesenchymal markers, such as N-cadherin and vimentin). Transcription factors that are upregulated in the EMT process include Twist, SNAI1 (snail), and SNAI2 (Slug) [242,243]. Lung tissue samples obtained from IPF patients demonstrate an increase in the expression of Twist and Snail, suggesting that EMT-associated signaling pathways are activated [244,245]. The balance between TGF-β and bone morphogenetic proteins (BMPs) is also important in the development of the mesodermal/epithelial compartment and in regulation of EMT. TGF-β is a potent inducer of EMT, whereas BMP antagonizes TGF-β-dependent signaling. Expression of Gremlin, an antagonist of BMP signaling, is upregulated in IPF lungs, indicating that the balance between TGF-β and BMP signaling is dysregulated [205,231,246,247].

### 4.3. IPF and UPR: Cellular Mechanisms 

#### 4.3.1. UPR Signaling in IPF

Emerging evidence implicates ER stress in the pathogenesis of IPF. Lung samples from both FIP and sporadic IPF patients have shown the presence of ER stress [93,155,248]. Strong positive staining for BiP, ERAD-enhancing α-mannosidase-like proteins (EDEM), and XBP1 in epithelial cells from IPF lungs, expression of misfolded precursor protein for surfactant protein-C (pro-SP-C), mutant protein aggregates in A549 type II alveolar epithelial cells, and expression of p50ATF6 (i.e. a processed form of ATF6), ATF4, and CHOP in sporadic IPF lung samples support a correlation between ER stress and IPF. Importantly, mutations in surfactant protein C (*SFTP*) C and A2 (*SFTPA2*) infer a role for ER stress in FIP [249,250]. Moreover, several cellular in vitro and animal in vivo studies suggest that increased ER stress is a key component of IPF pathogenesis [155,251,252,253,254,255,256,257,258,259]. A schematic summary of IPF pathology and the role of ER stress is shown in Figure 3. 

#### 4.3.2. Exogenous Stimuli Contributing to ER Stress in IPF

##### Smoke

Multiple in vitro and in vivo studies have confirmed that CS and other inhalational pollutants can trigger ER stress and the UPR through the active phosphorylation of eIF2α by PERK followed by the induction of ATF4, GADD34, and UPR markers, such as BiP, XBP1, and GRP94. Another mechanism through which cigarette smoking induces ER stress is by altering cellular redox status. CS contains many oxidizing agents that can impair the function of ER by affecting disulfide bond formation in PDI, a redox sensitive chaperone. Acrolein, hydroxyquinones, and peroxynitrite, which emanate from smoke, lead to PDI nitrosylation of cysteine and tyrosine residues and a reduction in disulfide bonds, resulting in the formation of misfolded proteins [260]. 

##### Viruses

Airborne viruses may be involved in the onset and progression of IPF. Implicated viruses include human herpes viruses (HHVs), a large family of ubiquitous DNA viruses, such as the common pathogen herpes simplex virus type 1 (HSV-1)), cytomegalovirus (CMV), Epstein–Barr virus (EBV), HHV-7, and HHV-8. HHV infection induces all three branches of UPR signaling (i.e. PERK, ATF6, and IRE-1) to favor its own replication. However, convincing evidence confirming that UPR activation by viruses leads to IPF is currently lacking. Another mechanism through which HHV may induce fibrosis is by causing a cytokine imbalance favoring IL-4, IL-1β, and TGF-β1 [261,262,263,264,265]. Coronaviruses might be also involved in the pathogenesis of IPF. Previous investigations showed respiratory coronavirus infections such as Severe Acute Respiratory Syndrome (SARS) and Middle East Respiratory Syndrome (MERS) have severe fibrotic consequences in the lung and increase the risk of developing fibrotic lung diseases including IPF [266]. These observations could be potentially true for Severe Acute Respiratory Syndrome Coronavirus 2 (SARS-CoV-2) infection which is responsible for the Coronavirus Disease 2019 (COVID-19) pandemic. A recent study showed that COVID-19 is associated with increased mortality in IPF patients [267].

##### UPR-Associated Diagnostic Markers in IPF

To date, no ER stress-induced UPR-specific diagnostic marker for IPF has been identified. However, the existence of key mediators of the three UPR arms in lung samples obtained from patients provides a predictable impression of IPF features induced by ER stress. Expression levels of BiP, EDEM, XBP1 (34–60), p50ATF6, ATF6, ATF4, CHOP, and misfolded pro-SP-C mutant protein aggregates in AECs-II can serve as potential diagnostic markers of ER stress in IPF [155,248,251,252]. Further research is needed in this area before reliable diagnostic markers can be developed. 

## 5. Conclusions and Future Directions

ER stress can be triggered by several common pathogenic mechanisms affecting the lungs. In recent years, the effects of ER stress and activated UPR have attracted enormous interest both as a cause and consequence in inflammation and fibrosis in several lung diseases including COPD, asthma, cystic fibrosis (CF), and IPF, respectively [265,268,269,270]. Continuous exposure of pulmonary cells to diverse environmental stimulants can activate UPR pathways specifically in the lung. In COPD, there is an increased expression of UPR transcription factors, such as p-eIF2α, CHOP, and several proteins involved in the ERAD pathway. Although the ATF6 and IRE1 UPR arms are not affected, expression of these UPR transcription factors has a positive correlation with the degree and severity of airflow obstruction [268,269,271]. In IPF, increased expression of BiP, XBP1, IRE1, ATF6, and CHOP in AECs-II, as well as the induction of ATF4, CHOP, and BAX, is evident [231,272,273]. Thus, both ER stress and UPR appear to be critically involved in several chronic and debilitating respiratory disorders. 

CS induces alterations in protein metabolism and the UPR cascade via oxidative stress, which irreversibly damages a variety of lung proteins. Acute exposure to CS causes an increase in the expression of BiP, calnexin, calreticulin, ATF4, PERK, p-eIF2α, and CHOP. In general, smoking not only increases the loading of proteins in the ER but also reduces the ability of ER capacity [265,274]. Overall, while several in vivo and in vitro studies have shown the profibrotic effects of ER stress, our understanding of the exact role of UPR in the pathogenesis of lung diseases remains limited. 

Targeting ER stress and UPR components may have therapeutic benefits in the treatment of lung diseases. For instance, in mouse models of lung fibrosis, the chemical chaperones 4-phenyl butyric acid (4-PBA) and taurohyodeoxycholic acid (TUDCA) showed beneficial effects [265,268,275]. In addition, the use of specific inhibitors of IRE1 could represent a potential therapeutic approach in IPF [93].

In recent years, there have been advancements in our understanding of the source of fibroblasts. It was previously thought that resident fibroblasts were the only source of these cells in different organs. However, fibrocytes have been identified as one of the sources for fibroblasts [276]. The detection of circulation fibrocytes could serve as an indicator and potential biomarker for IPF and COPD [277,278,279]. Fibrocytes are produced in the bone marrow stroma, and they transmigrate from the blood to sites of lung injury via different chemokine gradients [279]. They later change their phenotype to resident fibroblasts and participate in the organs’ response to stress. Therefore, targeting such fibrocyte pathways and their subsequent phenoconversion could be targeted for future therapeutic development in the treatment of IPF and COPD. Given that UPR is involved in the regulation of phenotype conversion [93] and cellular secretome [78], targeting these pathways in fibrocytes could be a new therapeutic strategy for both COPD and IPF. 

In this review, we have discussed the emerging role of ER stress and UPR activation in COPD and IPF. Research on the ER stress–UPR axis is in its infancy; therefore, future in-depth research efforts are warranted to further investigate this complex pathway and its impact on the onset and progression of chronic lung diseases, with the ultimate goal of developing novel therapies. Of note, UPR signaling pathways are common and required for the maintenance of physiological balance as well. Therefore, comprehensive in vitro and in vivo research approaches are required for the identification and development of specific and safe ER/UPR-modulating therapies.

Epigenetic regulation of ER stress is also of importance when considering disease therapy. An accumulating body of evidence demonstrates the regulation of ER stress by ncRNAs mediators such as miRNAs, lncRNAs, and circRNAs [280,281,282,283]. Various interactions between ncRNAs and ER stress mediators have already been identified in different types of cancers [284,285]. Future research endeavors are needed to elucidate the relationships between ncRNAs and the UPR in IPF and COPD. Such studies may reveal that targeted strategies to modulate aberrant ER stress and UPR signaling have promising therapeutic potential in the treatment of chronic lung diseases.

## Figures and Tables

**Figure 1 life-11-00001-f001:**
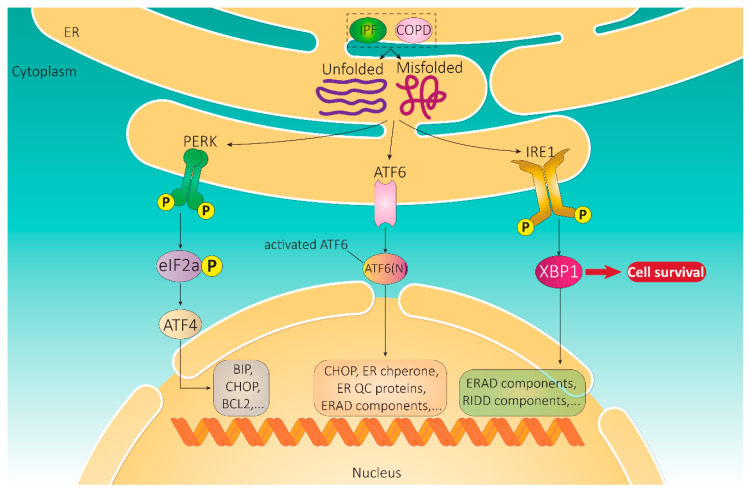
Schematic representation of the unfolded protein response (UPR) pathway. Idopathic pulmonary fibrosis (IPF) and chronic obstructive pulmonary disease (COPD)-involved mechanisms are generally involved in UPR induction. Incorrectly folded proteins bind to (UPR) sensor proteins in the endoplasmic reticulum (ER) lumen, resulting in activation of the UPR. Protein kinase R-like ER kinase (PERK), activating transcription factor 6 (ATF6), and serine/threonine-protein kinase/endoribonuclease inositol-requiring enzyme (IRE)-1 activate a series of reactions and signaling pathways, which eventually leads to transcription initiation and translation regulation of the effector genes. These genes include C/EBP-homologous protein (CHOP) and components of the ER-associated degradation (ERAD) system and regulated IRE1-dependent decay (RIDD), which regulate apoptosis/autophagy, ER expansion, and protein folding. Abbreviations: eIF2α (eukaryotic initiation factor 2 alpha), ATF4 (activating transcription factor 4), XBP-1 (X-box binding protein-1), ER chaperone-binding immunoglobulin protein (BiP).

**Figure 2 life-11-00001-f002:**
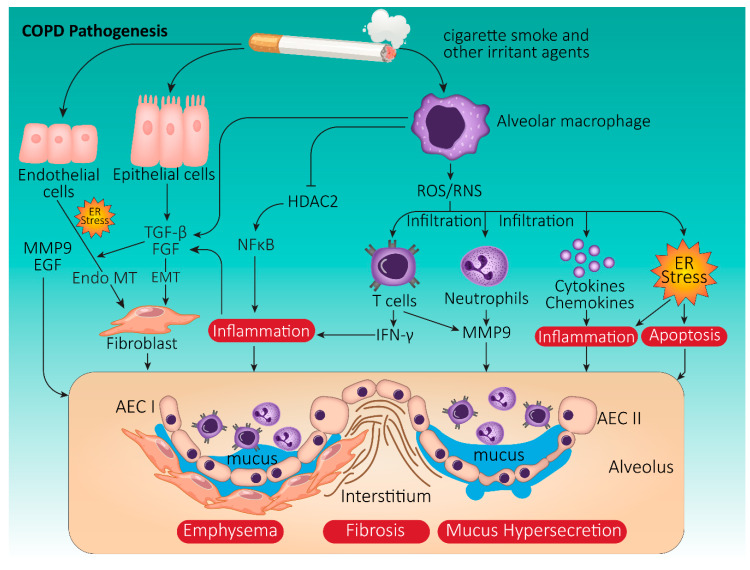
Schematic summary of COPD pathogenesis. Cigarette smoking is the most common risk factor triggering the development of chronic obstructive pulmonary disease (COPD). In response to irritant exposure, alveolar macrophage activation generates excessive reactive oxygen (ROS) and nitrogen (RNS) species, leading to T cell and neutrophil infiltration, cytokine/chemokine release, inflammation, and endoplasmic reticulum (ER) stress. Smoking damages resident epithelial cells in the lung, which promotes further release of pro-inflammatory factors, transforming growth factor-β (TGF-β), and matrix metalloproteinase-9 (MMP-9). All these factors trigger inflammation, alveolar epithelial cell (AEC) I and II apoptosis, fibrosis, and mucus hypersecretion, contributing to the development of airflow obstruction and emphysema. Abbreviations: FGF (fibroblast growth factor), HDAC2 (histone deacetylase 2), EGF (epidermal growth factor), INF-γ (interferon gamma).

**Figure 3 life-11-00001-f003:**
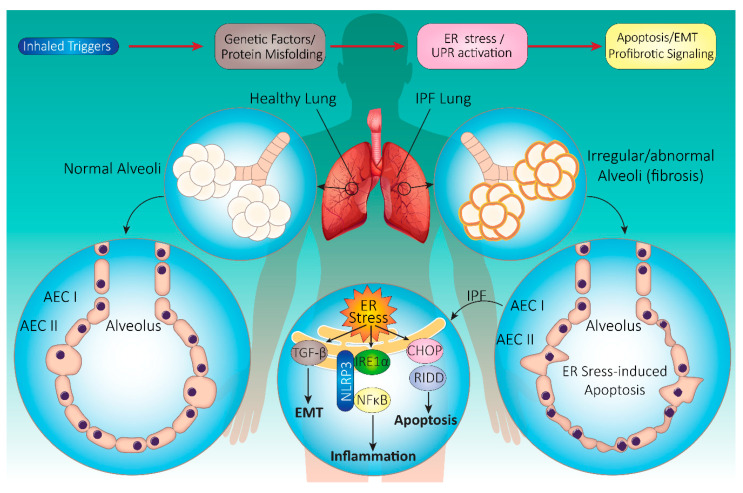
Schematic summary of IPF pathogenesis. Inhaled triggers along with genetic factors contribute to the induction of UPR in IPF pathogenesis. UPR activation triggers many downstream pathways, which may induce EMT, apoptosis, and pro-fibrotic signaling in the lung. The key histopathological feature of IPF is usually interstitial pneumonia (UIP). Endoplasmic reticulum (ER) stress in IPF triggers apoptosis through UPR system activation, inflammation via nuclear factor kappa-light-chain-enhancer of activated B cells (NFκB) activation, and epithelial–mesenchymal transition (EMT) through transforming growth factor-β (TGF-β) activation. Affected cells include alveolar epithelial cell (AEC) I and II, resident macrophages, and fibroblasts in the alveoli. Alveolar changes lead to exaggerated fibrosis, lymphocyte infiltration, and honeycomb change. Abbreviations: CHOP (C/EBP-homologous protein), IRE1 (serine/threonine-protein kinase/endoribonuclease inositol-requiring enzyme 1), RIDD (regulated IRE1-dependent decay), NLRP3 (NOD-, LRR-, and pyrin domain-containing protein 3).

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
