# Peer review of "The ER Stress/UPR Axis in Chronic Obstructive Pulmonary Disease and Idiopathic Pulmonary Fibrosis"

_life, 2020, doi:10.3390/life11010001_

Round 1

Reviewer 1 Report

In this review, Aghaei et al. described major routes for ER stress and UPR linked to the pathophysiology of IPF and COPD.

The manuscript is interesting, well written, with well-structured and informative figures.

Please find few suggestions below.

1) The introduction part should be reworked:

a) The two parts from line 64 and line 90 could be clarified.

b) I would suggest to keep in the introduction part only the two paragraphs of this chapter and to continue directly with chapter 2. The third and the forth paragraphs concerning IRE1 might be add to the IRE1 part from chapter 2. Informations regarding BIP and FIC proteins should be move elsewhere, after PERK for instance.

2) Since BIP is the name use in the main text, replace GRP78 by BIP in Figure 1 and in part 4.3.2.3.

3) Cite Figure 1, 2 and 3 earlier in the text (for example, cite figure 3 when finishing the IPF pathophysiology part)

4) The part of COPD is too big compared to the part of IPF. Authors could try to shorten the COPD part and to elaborate on the IPF part.

5) As the role of non-coding RNAs is only exposed in relation to the COPD pathophysiology it might be removed from the abstract

6) Figures are nice and informative, however it would have been more informative for the purpose of the review to have figures focusing on the UPR system within the diseases pathophysiology

Author Response

Comments and Suggestions for Authors

In this review, Aghaei et al. described major routes for ER stress and UPR linked to the pathophysiology of IPF and COPD.

The manuscript is interesting, well written, with well-structured and informative figures.

Please find few suggestions below.

1) The introduction part should be reworked:

  1. The two parts from line 64 and line 90 could be clarified.

Answer: We appreciate the reviewer feedback. We re-word line 64-68 so it reads easier. We also deleted line 74-90, which was related to IRE1 non-canonical pathway. As this part was not directly linked to our review paper and it makes the introduction very complex.

  1. I would suggest to keep in the introduction part only the two paragraphs of this chapter and to continue directly with chapter 2. The third and the forth paragraphs concerning IRE1 might be add to the IRE1 part from chapter 2. Informations regarding BIP and FIC proteins should be move elsewhere, after PERK for instance.

Answer: This is a very well thought comment. We moved the IRE1 related parts to 2.1.1 (lines:169-187). We also moved the parts related to BiP to section to just after dicussions related to BIP (lines: 88-118).  These changes made the revised version much fluent and we really appreciate the respected editor feedback.

2) Since BIP is the name use in the main text, replace GRP78 by BIP in Figure 1 and in part 4.3.2.3.

Answer: We have changed the GRP78 to BIP and modified Figure 1.

3) Cite Figure 1, 2 and 3 earlier in the text (for example, cite figure 3 when finishing the IPF pathophysiology part).

Answer: We have cited Figures in different parts of revised manuscript based on the respected reviewer feedback (Red font).

4) The part of COPD is too big compared to the part of IPF. Authors could try to shorten the COPD part and to elaborate on the IPF part.

Answer: We have summarized COPD part to balance with IPF part.

5) As the role of non-coding RNAs is only exposed in relation to the COPD pathophysiology it might be removed from the abstract.

Answer: We added “COPD” after non-coding RNAs in abstract to indicate it is only discussed in COPD.

6) Figures are nice and informative, however it would have been more informative for the purpose of the review to have figures focusing on the UPR system within the diseases pathophysiology

Answer: We added some new part in Figure1),  Figure 2 and 3 to indicate the role of UPR in COPD and IPF pathogenesis.

Reviewer 2 Report

The manuscript concerns the mechanistic links between ER stress, the unfolded protein response in COPD and IPF. The review covers the main pathways and molecular elements of the ER stress/UFP axis. Figures are well constructed, and generally speaking the text is concise but accessible to readers new to the field.

The authors should consider the following suggestions for revision.

1) Reference to and integration of figures.
Throughout the text the figures are referred to only once each however, they contain elements and conceptual relationships that are frequently mentioned in the text. It is suggested that the authors make more reference to the figures throughout the text, particularly in the earlier parts of each section during exposition of relatively well-characterised concepts. The first figure in particular could be moved to an earlier page in the manuscript as it gives a basic overview of the UPR pathways.

2) Introduction
The introduction would benefit from a few subheadings in order to break up the text and indicate the theme of each set of paragraphs. The sequence of subheadings should be the logical progression of concepts and information the authors wish to convey.

The second paragraph of the introduction (Line 64 to 73) would benefit from re-writing.
Suggested changes to the first half of the 2nd paragraph of the introduction (* indicates change) -

"The UPR signalling pathway in ER stress was first identified in Saccharomyces, in which ER stress
is exclusively controlled by the serine/threonine-protein kinase/endoribonuclease inositol-requiring
enzyme 1 (IRE1). This is in contrast to vertebraTeS*, where UPR is regulated by three main signalling arms:* IRE1, activating transcription factor 6 (ATF6) β, and protein kinase R like ER kinase
(PERK) (see Figure 1)*. "

The second half of the 2nd paragraph of the introduction is overly concise, and would benefit from a moderate expansion.

3) Fibrocytes
The authors should briefly mention or speculate on the role played by circulating fibrocytes in IPF or lung fibrosis associated with chronic inflammatory disease, such as COPD.

4) Coronavirus and IPF
The authors should take the opportunity to mention the potential of novel coronaviruses such as MERS or SARS to cause pulmonary fibrosis in section 4.3.2.2 (Viruses).

5) Line 435 - (PMID: 18079489; PMID: 28464871) should be replaced by references.
Line 442 - (PMID: 24111704) should be replaced by a reference.

Author Response

Comments and Suggestions for Authors

The manuscript concerns the mechanistic links between ER stress, the unfolded protein response in COPD and IPF. The review covers the main pathways and molecular elements of the ER stress/UFP axis. Figures are well constructed, and generally speaking the text is concise but accessible to readers new to the field.

The authors should consider the following suggestions for revision.

1) Reference to and integration of figures.

Throughout the text the figures are referred to only once each however, they contain elements and conceptual relationships that are frequently mentioned in the text. It is suggested that the authors make more reference to the figures throughout the text, particularly in the earlier parts of each section during exposition of relatively well-characterized concepts. The first figure in particular could be moved to an earlier page in the manuscript as it gives a basic overview of the UPR pathways.

Answer: We have followed the respected reviewer point and referred more to the Figures in the revised version (red font).

2) Introduction

The introduction would benefit from a few subheadings in order to break up the text and indicate the theme of each set of paragraphs. The sequence of subheadings should be the logical progression of concepts and information the authors wish to convey.

Answer: We have relocated some parts of “Introduction” to other realted parts. For example we moved the parts which are related to IRE1 to “IRE1 sub-heading, 2.1.1” and general parts related to UPR/BIP to the “UPR, 2” sub-heading.

The second paragraph of the introduction (Line 64 to 73) would benefit from re-writing.

Answer: We appreciate the reviewer feedback. We re-word line 64-68 so it reads easier. We also deleted line 74-90, which was related to IRE1 non-canonical pathway. As this part was not directly linked to our review paper and it makes the introduction very complex.

Suggested changes to the first half of the 2nd paragraph of the introduction (* indicates change) -

"The UPR signalling pathway in ER stress was first identified in Saccharomyces, in which ER stress is exclusively controlled by the serine/threonine-protein kinase/endoribonuclease inositol-requiring enzyme 1 (IRE1). This is in contrast to vertebraTeS*, where UPR is regulated by three main signalling arms:* IRE1, activating transcription factor 6 (ATF6) β, and protein kinase R like ER kinase (PERK) (see Figure 1)*. "

Answer: We applied the suggestion to mentioned paragraph (lines: 64-68) in the revised version.

The second half of the 2nd paragraph of the introduction is overly concise, and would benefit from a moderate expansion.

Answer: We appreciate the respected reviewer comments. As the other reviewer asked us to summarize the introduction and relocate some of the parts, we did a strategy to fulfill both reviewer recommendation. We moved the second part of the inoruction to two different parts of of manuscript. We moved the parts related to IRE1, to the end of the 2.1.1 section (lines: 169-187) and the part related to BIP and the part related to FIC and BIP to the sub-section 2 (lines: 88-118). This strategy help to improve the flow the manuscript and make it for easier to follow for the readers.

3) Fibrocytes

The authors should briefly mention or speculate on the role played by circulating fibrocytes in IPF or lung fibrosis associated with chronic inflammatory disease, such as COPD.

Answer: We have added a paragraph about the role of “fibrocytes” in both COPD and IPF in the conclusion section and suggested potential application of UPR in developing of new treatment in these diseases via targeting the fibrocytes (lines: 637-648).

4) Coronavirus and IPF

The authors should take the opportunity to mention the potential of novel coronaviruses such as MERS or SARS to cause pulmonary fibrosis in section 4.3.2.2 (Viruses).

Answer: We added a paragraph in section 4.3.2.2 to address the potential importance of coronavirus infection in IPF (lines: 600-606).

5) Line 435 - (PMID: 18079489; PMID: 28464871) should be replaced by references.

Line 442 - (PMID: 24111704) should be replaced by a reference.

Answer: Thank you for the note. We have replaced the references with PMIDs.

Round 2

Reviewer 1 Report

The authors really improved their review.